# Impedance Spectroscopy of Pr-Doped BaBi_2_Nb_2_O_9_ Aurivillius Ceramics

**DOI:** 10.3390/ma15186308

**Published:** 2022-09-11

**Authors:** Michał Rerak, Jolanta Makowska, Małgorzata Adamczyk-Habrajska, Lucjan Kozielski

**Affiliations:** Faculty of Science and Technology, University of Silesia, 1A 75 Pułku Piechoty St., 41-500 Chorzów, Poland

**Keywords:** ceramics, Aurivillius structure, BaBi_2_Nb_2_O_9_, praseodymium Pr^3+^, impedance spectroscopy

## Abstract

Herein this study, the polycrystalline nature of the Aurivillius type structure is studied; primarily, the main objective is to observe the influence of dopant Pr^3+^ at the Ba^2+^-site of BaBi_2_Nb_2_O_9_ (BBN) ceramics. The ceramics under investigation were fabricated via the conventional solid-state reaction method. Scanning electron microscopy (SEM) and energy dispersion spectroscopy (EDS) techniques were used to analyse their morphological structure. It was found that the chemical composition of the ceramic samples corresponds well to the initial stoichiometry of the ceramic powders. An increase in praseodymium content caused a slight decrease in the average size of the ceramic grains. The obtained ceramic materials are described by a tetragonal structure with the space group I4/mmm. The electrical properties of the material have been studied using complex impedance spectroscopy methods in wide temperature and frequency ranges. The analysis of obtained results showed grains and grain boundaries contribute to conductive processes in the material. A possible ’hopping’ mechanism for electrical transport processes in the system is evident from the analysis of results based on Joncher law.

## 1. Introduction

In the ideal form, the crystal structure of ABX_3_ perovskite materials can be described as consisting of corner-sharing [BX_6_] octahedral with A cation occupying the 12-fold coordination site formed in the middle of the cube of eight such octahedral [1,2]. This cell is far from ideal because its deformations result from stresses in the structure [3]. They result from the difference in the ionic radii of the elements that make up a given compound. It is these deformations that determine the physical properties of the material. The commonly known representatives of perovskite electroceramics are PbZrO_3_-PbTiO_3_ [4,5] or PbFe_1/2_Nb_1/2_O_3_ [6,7]. The structure of the perovskite is the starting form for the construction of compounds with the structure of layered perovskites; such structures include the Aurivillius structures (Bi-layered structures). These structures have the following general formula [Bi_2_O_2_][A_n−1_BnO_3n+3_] (n = 1, 2, 3, 4) where [Bi_2_O_2_]_2+_ layers [8] are interleaved with n perovskite—type layers having composition [A_n−1_B_n_O_3n+1_]^2−^ [9]. The studies in recent years focused on the high-temperature piezoelectric properties for higher members in the series (n ≥ 2), such as CaBi_4_Ti_4_O_15_ [10], CaBi_2_Nb_2_O_9_ [11,12], and Bi_3_TaTiO_9_ [13].

Another representative of this group of compounds is BaBi_2_Nb_2_O_9_ (BBN) ceramics, recognized as important material for microelectronics and non-volatile random access memory devices application [14,15,16,17]. The material is characterized by a relatively small value of electric permittivity at room temperature compared to lead-based ceramics. [18,19,20]. The value of the electric permittivity increases with temperature and achieves its maximum at T_m_ temperature. In addition, the discussed material shows features characteristic of ferroelectric relaxors, namely the strong frequency dispersion of temperature T_m_ and maximum value of permittivity [16]. The features are connected with the presence in the tetragonal matrix of polar nano regions with orthorhombic distortion [21]. The polar nano-regions are tailored by the incorrect distribution of Ba ions in the crystal lattice. Ismunandar et al. [22], based on XRD measurements, have proven that Ba ions entered not only the perovskite blocks but also (Bi_2_O_2_)^2+^ layers resulting in the inhomogeneous distribution of Ba ions and local charge imbalance in the layered structure [23].

It was found that the trivalent rare ions doping Sm^3+^ or La^3+^ the Bi^3+^ in BBN and SrBi_2_Nb_2_O_9_ (SBN) ceramics influence the diffuseness of ε(T) characteristic and also electrical conductivity [24,25,26,27]. In light of the presented facts, substituting rare trivalent ions in the place of Ba^2+^ ions as a heterovalent dopant seems interesting. Taking into account the reports of other authors on the positive effect of praseodymium Pr^3+^ ions on the luminescent properties of compounds from the Aurivillius phase group [28] and not forgetting about the application aspect of the research, the authors of this study decided to dope BBN ceramics with praseodymium ions. The use of an appropriate chemical formula, also described in our previous article [29], forced the introduction of praseodymium ions in place of barium. The obtained results seem to confirm the assumed stoichiometry. Namely, the influence of praseodymium modification on crystal structure is considerable. However, the microstructure and dielectric properties change significantly and the ferroelectric relaxor features of BBN ceramics. All mentioned features are widely described in our previous paper [29]. Praseodymium admixture causes an increase in the value of diffuseness coefficient and frequency dispersion of emax and T_m_. The results are in contrast to those obtained by the authors of [30,31] for SBN ceramics. Namely, the introduction of praseodymium ions in place of Bi^3+^ ions resulted in a decline in the diffuseness of maximum of ε(T) dependences and a reduction in the frequency dispersion of ε_max_ and T_m_. 

Considering the potential application possibilities of the modified BBN ceramics, praseodymium ions’ influence on the obtained material’s electrical properties seems extremely interesting. This article broadly describes the effect of praseodymium admixture on thermally stimulated depolarization currents, which are connected with the space charge density. Moreover, the results of impedance spectroscopy measurements of the discussed materials were analyzed in detail. Based on this analysis, the influence of praseodymium ions on the conductivity mechanism occurring in the discussed materials has been discussed.

## 2. Materials and Methods

The preparation and synthesis process of undoped and Pr^3+^ co-doped BBN powders were performed through the solid-state reaction, by the conventional ceramic method. Stoichiometric amounts of high purity powders, niobium oxide (V) Nb_2_O_5_ (Aldrich 99.9%), bismuth oxide (III) Bi_2_O_3_ (Aldrich 99.9%), barium (III) carbonate BaCO_3_ (Aldrich 99.9%), and praseodymium oxide (III) Pr_2_O_3_ (POCH, CZDA) were weighed according to the nominal composition of BPBN. The appropriate quantities of reagents were weighed according to the formula (1):(1)$$\left(2-2x\right){\mathrm{BaCO}}_{3}+x{\mathrm{Pr}}_{2}O_{3}+2{\mathrm{Bi}}_{2}O_{3}+2{\mathrm{Nb}}_{2}O_{5}\to 2{\mathrm{Ba}}_{1-x}{\mathrm{Pr}}_{x}{\mathrm{Bi}}_{2}{\mathrm{Nb}}_{2}O_{9+x/2}+\left(2-2x\right){\mathrm{CO}}_{2}↑$$

The powders were first mixed, pressed into pallets of d = 25 mm in diameter and h = 2 mm thick at p = 300 MPa, and calcined at temperature T = 950 °C for t = 4 h. Next, the material was milled in a wet medium and dried. The synthesized ceramic powder was then pressed and formed into pellets under the pressure of p = 600 MPa. BPBN ceramics were sintered in air at temperature T = 1100 °C for 2 h. The flowchart of the complete fabrication process is shown in Figure 1.

Microstructure and analysis of chemical composition by EDS method of BBN ceramic samples of reference composition and doped with praseodymium were made based on a JEOL JSM-7100F TTL LV scanning electron microscope (Akishima, Japan) equipped with a JEOL-EDXS X-ray energy dispersion spectrometer with an energy resolution of 138 eV.

The X-ray tests were performed at room temperature on a Philips X’pert diffractometer (Amsterdam, The Netherlands) with a CuKα1 and α2 radiation (40 kV, 30 mA) with step 0.04◦ and time adjusted to receive accurate counting statistics. Computations with obtained data were carried out using the Rietveld method [32] using the LHPM computer program (version 4.2.).

Measurements of thermally stimulated depolarization currents were performed for poled ceramic samples. Before measurements, the samples were first polarized at dc field with strength Ep = 1 kV applied for 10 min at temperature Tp = 373 K and then cooled in the field to 273 K, at which the field was switched off. The samples were then heated at constant rate of 5 K/min to the temperature of 723 K. The depolarization current was measured with a picoammeter (Keithley 6485, Cleveland, OH, USA).

The measuring system for impedance spectroscopy tests consisted of a Hewlett-Packard type 4192 A impedance analyzer, a Hewlett-Packard type 34401 A millivoltmeter (Palo Alto, CA, USA), and a Shimaden FP93 temperature controller. The impedance measurements were made in the temperature range T = 500–823 K in steps of T = 10 K in the frequency range of the measurement field f = 20 Hz–2 MHz. Before analyzing the results, a test was performed each time, which allowed determining the consistency of the data using the Kramers–Kroning relationship.

## 3. Results and Discussion

In order to check the influence of the praseodymium crystal structure of BBN ceramics, the XRD measurements were performed. The obtained X-ray diffraction pattern is presented in Figure 2.

The obtained results, namely the intensity and position of the diffraction lines, were compared with the one from the international database (JCPDS standard number 12-0403). The analysis indicated that the discussed ceramic materials are single-phase at room temperature. The praseodymium ions do not influence the crystal structure. All discussed materials, regardless of the concentration of the dopant ions, are characterized by the tetragonal phase with the I4/mmm space group. The lattice parameters were determined using the Rietveld method. The method, as well as the results of the analysis, were widely described in our previous paper [29]. The admixture of praseodymium had little effect on the parameters of the crystal lattice. The unit cell volume equals 398.1 Å^3^ and 395.8Å^3^ for pure BBN ceramics and *x* = 0.1, respectively. The calculated values were used to determine the theoretical density of discussed materials. This density was compared with the experimental one measured by the Archimedean method. The experimental density was from 93 to 98% of the theoretical density [29], which allows expecting dense structures with a small number of pores. The microstructural tests conducted with the use of a scanning electron microscope confirmed these assumptions (Figure 3).

The images of the microstructure of pure and praseodymium admixture BBN materials show the well-developed grains and grains boundaries. Grains have a plate-like shape with rounded corners, which results from the high grain growth in the direction perpendicular to the c-axis. Such manner of grains’ growth is the reason for the anisotropic nature of Aurivillius materials’ crystal structure [33,34]. The image of pure BBN ceramics’ microstructure stays in good agreement with the one described by Sunanda K. Patri et al. [35]. The grains’ size slightly decreases with an increase in dopant content. The average grain size changes from ca. 2 µm [36] to ca. 1.4 µm [29], for undoped and *x* = 0.1 doped ceramics, respectively. The slight grain refinement entails an increase in the homogeneity of the microstructure. The results are consistent with those obtained in the case of the modification by praseodymium ions of another representative of the Aurivillius phases, namely SBN ceramics [30,37]

The stoichiometry of obtained ceramics was investigated by EDS method. The study was performed in 50 randomly selected fracture micro-areas of each discussed ceramic material. Exemplary EDS spectra of the discussed materials are presented in Figure 3. The EDS analysis showed no deviations from the nominal compositions within the experimental error of the method, which confirms the formation of a solid solution with the assumed stoichiometry. Moreover, the analysis confirmed a homogeneous distribution of all the elements throughout the grains. The detailed results of the EDS analysis have been described in detail in our previous work [29].

The studies of the thermally stimulated depolarization currents have clearly shown that the space charge undoubtedly plays a significant role in shaping the electrical properties of pure BBN ceramics [35]. This type of charge is related mainly to the presence of oxygen vacancies. Modifying the base material by praseodymium admixture according to formula ${\mathrm{Ba}}_{1-x}{\mathrm{Pr}}_{x}{\mathrm{Bi}}_{2}{\mathrm{Nb}}_{2}O_{9+x/2}$ provided an increase in the amount of oxygen. Such action ensures the electroneutrality of the material while reducing the creation of oxygen vacancies. Reducing the concentration of oxygen vacancies should significantly affect the space charge density and thus the electrical properties of the ceramic material in question. The measurements of thermally stimulated depolarization currents have been made to confirm the thesis. The temperature changes of the observed TSDC are shown in Figure 4.

The maximum value of TSD currents declines with the increase of praseodymium content which is probably associated with the smaller number of oxygen vacancies.

Changes in the concentration of oxygen vacancies should affect the mechanism of electrical conductivity. In order to verify this thesis, the discussed ceramic materials were analyzed through impedance spectroscopy. The study was done in a wide range of temperatures (500–823 K) and frequencies (20 Hz–2 MHz). Figure 5 shows the dependence of the real and imaginary parts of impedance as a function of the frequency of the measurement field.

The shape of the logZ′ (logf) relationship (Figure 5) indicates that in the area of low frequencies, the value of the real part of the impedance decreases with increasing temperature which can be explained by the increase in electrical conductivity [38]. Moreover, the value of Z′ decreases with increasing frequency. On the other hand, the course of the logZ″(logf) relationship indicates that the value of the imaginary part of the impedance initially increases with the increase in frequency, reaching the maximum value of Z″_max_ for the frequency f_max_, and then gradually decreases. Additionally, the presented dependencies indicate that the values of Z″_max_ decrease with the increase of temperature, which is undoubtedly related to the cause increase in temperature, lowering the resistance of the material. It is also worth emphasizing that the half-width of the logZ″ (logf) curve is greater than 1.2 decades, proving that the properties of the materials in question deviate from the ideal Debay behaviour. The merging of the logZ″ (logf) characteristics into one curve, observed in the high-frequency range, indicates a large accumulation of space charge [39,40]. According to the authors of the work [41], the maxima observed on the logZ″ (logf) relationships are related to the existence of spatial charge relaxation, characteristic for materials consisting of grains and grain boundaries. On the other hand, the shift of the f_max_ frequency towards higher values with increasing temperature can be explained by the hopping model of conductivity present in this material—the rate of shift of the charge carriers increases with temperature, which significantly reduces the relaxation time of mobile charges [42].

In order to verify the consistency of the experimental data, all impedance data were subjected to a Kramers−Kroning (K−K) test using the software developed by Boukamp. [43,44,45]. Results of relative residuals for both the real and imaginary parts of the impedance for characteristics of all discussed ceramics measured at 700 K, as an example, are given in Figure 6.

Successive residual points are arranged randomly in relation to the frequency axis, their absolute value does not exceed 0.5%, and the coefficient χ^2^ ranges from 1.2 × 10^−8^ to 1.1 × 10^−6^. These facts clearly indicate that the obtained experimental data are consistent. Similar relationships were obtained for the remaining impedance spectra measured in the temperature range from T = (500–823) K. The positive data consistency test allowed for further steps to be taken to the analysis of the obtained impedance data. In the next step, the dependencies of the parameter Z/Z″_max_ (the so-called normalized part of the imaginary impedance) were plotted against the normalized frequency (f/f_max_). The characteristics presented in Figure 7 show that the aforementioned half-width decreases with increasing temperature for compositions with low praseodymium content (up to the admixture concentration *x* = 0.06).

The half-width of compositions with a higher content of admixture is practically independent of the temperature which suggests that the relaxation processes taking place in the material are independent of the temperature, resulting in the occurrence of one mechanism responsible for these processes [46,47,48]. Moreover, it is worth noting that the dependence Z/Z″_max_ (f/f_max_) merges into one curve in the high frequency range which confirms the large accumulation of space charge in the material [49].

Plotting the Nyquist dependence, i.e., the characteristics of the imaginary part of the impedance as a function of its real part, makes selecting an appropriate substitute electrical system much more effortless. This type of characteristic for selected temperatures is presented in Figure 8.

The shapes of dependencies presented in Figure 8 differ significantly from those proposed by Debye. This is due to the fact that the Debye model assumes the existence of one relaxation process characterized by an appropriate relaxation time. This fact entails the presence of a perfectly centred semicircle on the Z′ axis. The semicircles visible in the discussed figure are clearly deformed, and their centres are significantly shifted below the axis corresponding to the real part of the impedance—this is a consequence of the overlapping of two relaxation phenomena characterised by different relaxation times [50]. These phenomena are related to two microstructure components, namely grains and grain boundaries. Consequently, an electrical circuit consisting of two classic Voigt elements connected in series was selected to describe the electrical properties. The first element represents the electrical properties of the grains’ interior, while the second describes the grain boundaries. This approach to describing the impedance of materials in which both grains and their boundaries participate in conductivity is known as the brick layer model [51]. An additional advantage in favour of using this particular model is the shape of the characteristics of the imaginary part of the impedance and the electric modulus as a function of frequency. Figure 9 shows an example of the frequency characteristics of the imaginary part of the impedance and the electric modulus corresponding to the temperature.

All the presented curves are characterized by a distinct maximum. However, the frequency corresponding to the maximums related to the relationship Z″ (f) lies below the frequency corresponding to the maximums M″ (f). The separation of frequencies corresponding to the maxima M″ and Z″ indicates that the relaxation process is dominated by the transport of charge carriers over short distances and does not meet the requirements characteristic of the behaviour described by the Debye model [49,52,53,54]. Moreover, such behaviour excludes the occurrence of one relaxation process in the material [41,55]. 

In the next step of analyzing the results of impedance spectroscopy measurements, the process of matching the equivalent circuit to the results obtained was started. First, efforts were made to adjust the measurement data to a substitute circuit containing two classic Voigt elements (Figure 10). The obtained matches were characterized by a very high value of the χ^2^ parameter, which indicated their poor quality.

Therefore, according to the procedures described in the literature [50,56], the capacities in the equivalent circuit were replaced with constant-phase CPE elements. The quality of the fit had improved but was not yet satisfactory. The next step in selecting the electrical equivalent system was to add capacity connected in parallel to the branch describing the properties of grains (Figure 11).

The circuit modified in this way was characterized by the lowest value of the χ^2^ parameter which indicates a good fit quality. This fact influenced the decision to use this system for further analysis of impedance results, which allowed determining the capacity and resistance of the grains and the grain boundaries. Table 1 presents exemplary values of parameters describing the components of the considered electrical substitute systems at two selected temperatures.

The observed increase in the value of grain and grain boundary resistivity stays in good correlation with the decrease in the maximum of thermally stimulated depolarization current described above (Figure 4). Both facts seem to confirm the gradual reduction in the number of oxygen vacancies due to the admixture of praseodymium.

The obtained resistances of grains and grain boundaries at particular temperatures allowed us to determine the dependencies ln_RG_ (1/T) and ln_RGB_ (1/T) (Figure 12 and Figure 13).

The linear nature of the relationship indicates the activating form of conductivity processes. Based on the Arrhenius relationship (2), the activation energy of the conductivity process of both *E_G_* grains and *E_GB_* grain boundaries was determined (Table 2).
(2)$$R=R_{o}\exp\left(\frac{-E_{a}}{kT}\right)$$
where:

*E*_a_—activation energy of conductivity process,

*R_o_*—pre-exposure factor,

*k*—Boltzman constant.

The *E_G_* activation energy of the grain conductivity process for unmodified BBN ceramics is *E_G_* =1 eV [26]. Increasing the concentration of praseodymium raises this value. On the other hand, the value of the conductivity activation energy of grain boundaries drops from *E_GB_* = 0.98 eV for pure BBN ceramics [49] to *E_GB_* = 0.85 eV for the praseodymium-modified BBN ceramics with a mole fraction x = 0.10. In the case of pure BBN ceramics, the activation energy values of the conductivity process in the grains and grain boundaries are similar, which suggests that the electrical contact of the two discussed components of the microstructure is almost continuous (homogeneous) [16,35,36,57]. The difference between the values of the two discussed energies becomes visible in the case of the lowest considered concentration of praseodymium—the activation energy associated with the grains increases while the energy associated with the grain boundaries decreases. The observed differences deepen with the increase in praseodymium content, indicating that the conductivity process in ceramics containing a high concentration of admixture is mainly due to grain boundaries, i.e., places where various types of defects and structure imperfections accumulate.

In the next step, using the Formula (3) [24], the frequency dependencies of conductivity at selected temperatures were determined (Figure 14):(3)$$\sigma_{AC}=\sigma_{DC}+A\omega^{n}\;\;0<n<1$$
where:

*σ_AC_*—total conductivity,

*σ*_0_—frequency constant,

*A*—coefficient dependent on polarization and temperature,

*n*—a factor describing the degree of interaction between the charge carriers and the crystal lattice.

Two regions can be distinguished in the frequency dependence of conductivity, presented in Figure 14. The first one corresponds to the low-frequency range—the conductivity in this range is practically constant. Thus, it can be said that it is dominated by DC conductivity which is mainly related to the shift of charge carriers. When a certain cutoff frequency f_H_ is reached, dispersion appears due to the AC conductivity observed in the high-frequency range. The said cutoff frequency is called the hopping frequency. It is associated with the presence of the so-called conduction hopping mechanism. In this situation, the frequency dependence of conductivity can be described by Jonscher’s law. Adjusting the obtained results σ(f) to the above, the law made it possible to determine the value of direct current conductivity and the exponent n. The dependence of the natural logarithm from the DC current part of electrical conductivity on the reciprocal of temperature is linear, which allowed to apply the classical Arrhenius law and to determine the activation energy of the DC conductivity process. The values of the determined energy are given in Figure 15. For low praseodymium contents, the values of the determined energies are close to the values of the activation energy of the grain boundary conductivity process, which may suggest that the main burden of conductivity, in this case, rests precisely on this microstructure element. A further increase in the modifier content causes an increase in the value of the activation energy of DC conductivity, suggesting a greater inclusion of grain areas in the process.

At this point, we cannot forget about the exponent n, determined from Joncher’s law, which plays a significant role in the process of determining the conductivity mechanism. In the case of praseodymium-doped ceramics, the exponent n decreases with increasing temperature (Figure 16), which indicates that according to the correlated barrier hopping (CBH) model, the Coulomb potential wells overlap. Such superimposition of the potential well reduces the effective barrier height and facilitates the jump of a single electron between adjacent positions [58,59,60]. A deviation from the described behaviour is the temperature dependencies of the exponent n for ceramics doped with praseodymium with a concentration of *x* = 0.02 and *x* = 0.10. Above the temperature *T* = 700 K in these materials, the exponent n oscillates around a constant value.

## 4. Conclusions

The Aurivillius phase ceramics of ${\mathrm{Ba}}_{1-x}{\mathrm{Pr}}_{x}{\mathrm{Bi}}_{2}{\mathrm{Nb}}_{2}O_{9+x/2}$ with varying *x* (*x* = 0, 0.02, 0.04, 0.06, 0.08, 0.1) were prepared using the solid-phase synthesis and free sintering in air. All obtained materials show a tetragonal structure with the I4/mmm space group. The praseodymium admixture does not significantly influence the crystal lattice parameters. The microstructure of discussed ceramics consists of well developed, clear grains and distinct grain boundaries. The grain size changed from ca. 2 µm to ca. 1.4 µm for undoped and *x* = 0.1 doped ceramics, respectively. The compositional homogeneity was checked and found satisfactory from EDS analysis.

The used formula ${\mathrm{Ba}}_{1-x}{\mathrm{Pr}}_{x}{\mathrm{Bi}}_{2}{\mathrm{Nb}}_{2}O_{9+x/2}$ provided an increase in the amount of oxygen. Such an approach ensures the electroneutrality of the materials and reduces the number of oxygen vacancies. The assumed method of introducing praseodymium ions in place of barium ions was confirmed by the decreasing value of the maximum value of the thermally stimulated depolarization currents, observed for the increasing concentration of dopant ions. Thus, it can be assumed that the discussed substitution leads to a reduction in material defects. The reduction of the space charge concentration with the increase in the content of praseodymium ions increases the grain resistance and grain boundaries. Linearity of the dependence of the natural logarithm of both resistivity and the reciprocal temperature point at the activation character of the conductivity process in grains and grain boundaries.

## Figures and Tables

**Figure 1 materials-15-06308-f001:**
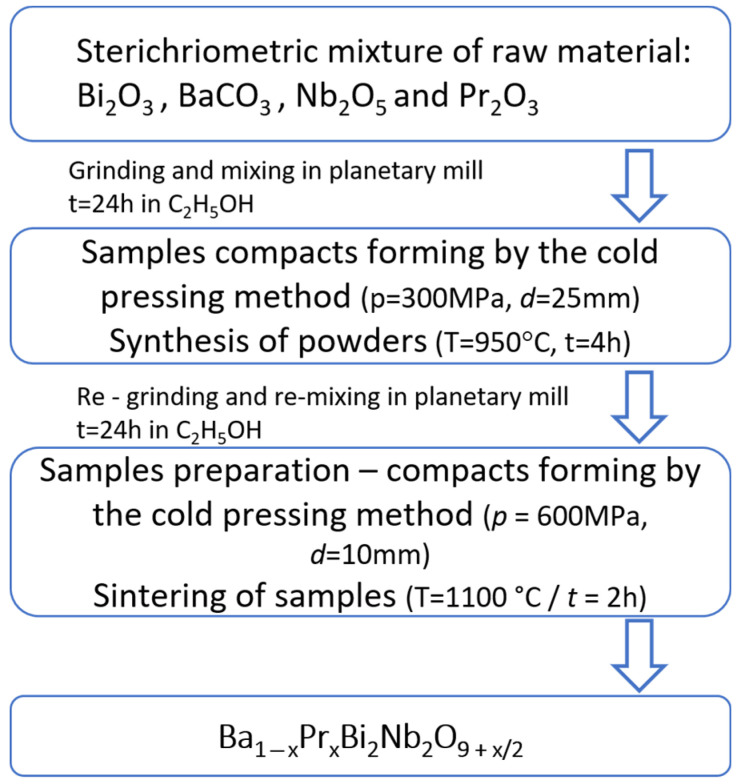
The flowchart of the fabrication process of BPBN ceramics.

**Figure 2 materials-15-06308-f002:**
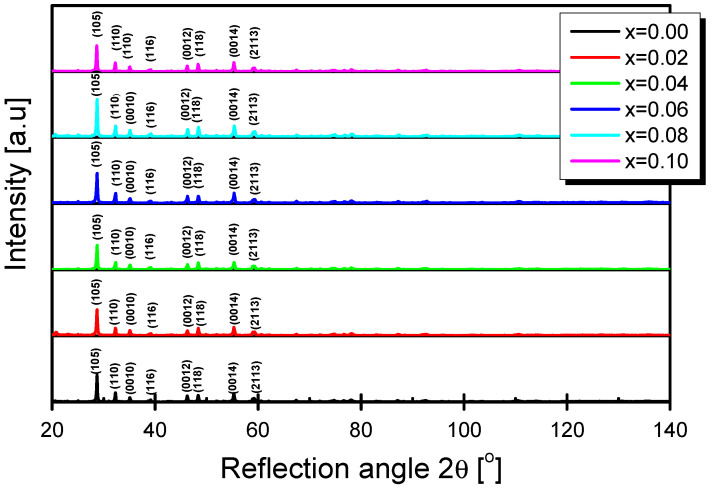
X-ray diffraction pattern of BBN ceramics doped with Pr^3+^ ions for various modifier concentrations.

**Figure 3 materials-15-06308-f003:**
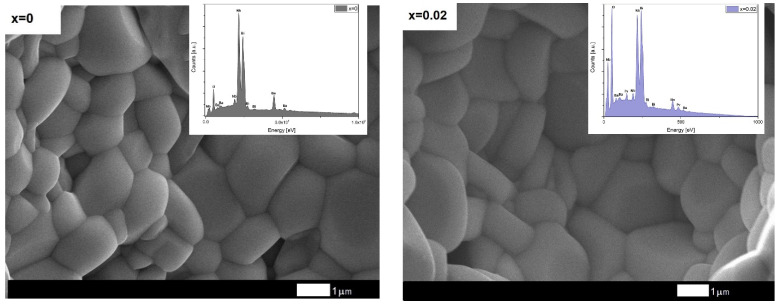
EDS spectrum and SEM photographs of fracture of BBN ceramics doped with Pr^3+^ ions.

**Figure 4 materials-15-06308-f004:**
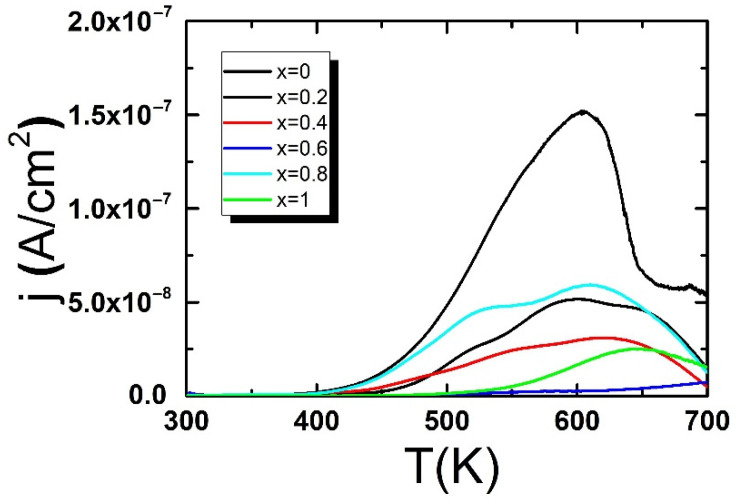
Thermally stimulated depolarization currents versus temperature for BBN ceramics.

**Figure 5 materials-15-06308-f005:**
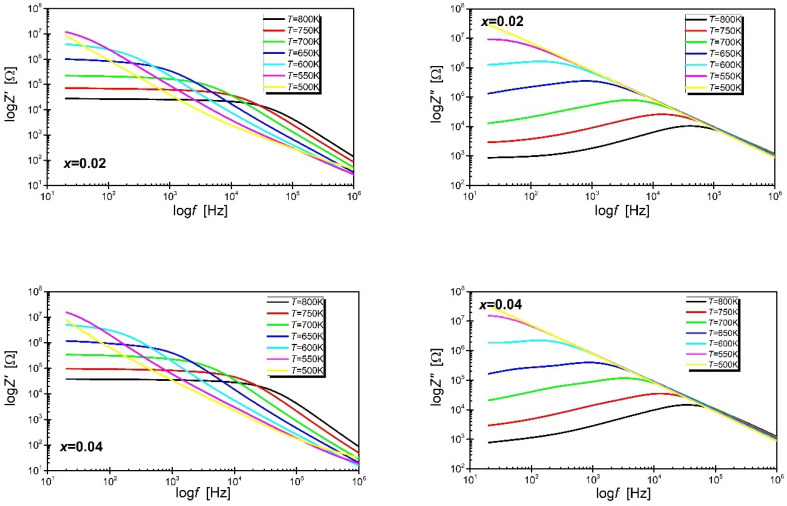
The frequency dependence of the real and imaginary impedance component of BBN ceramics modified with praseodymium ions was measured at selected temperatures in the range T = 500 K–800 K.

**Figure 6 materials-15-06308-f006:**
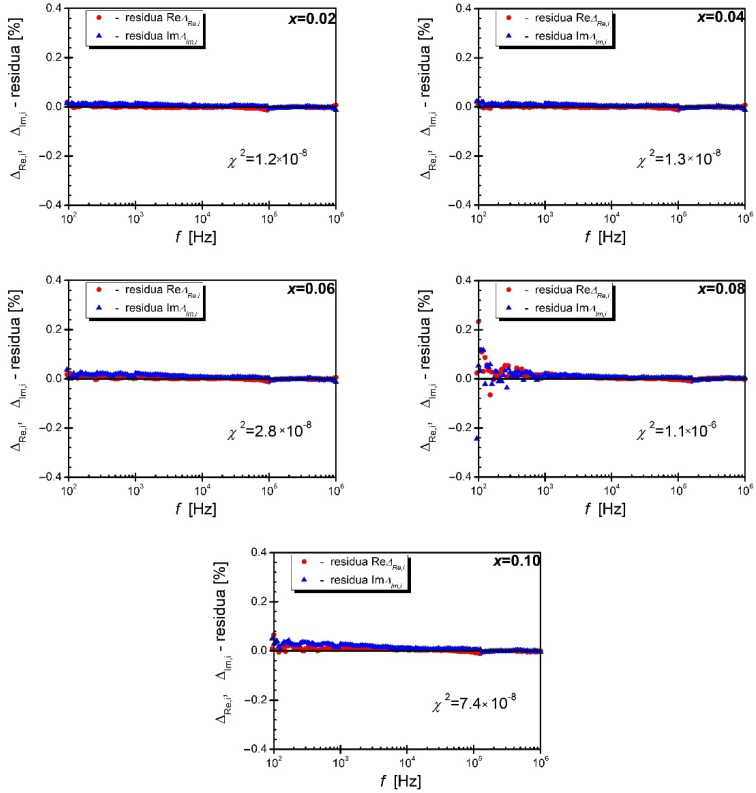
The frequency dependence of the real part residuals and the imaginary impedance of the BBN_9_ ceramics doped with praseodymium for mole fractions: *x* = 0.02, *x* = 0.04, *x* = 0.06, *x* = 0.08, and *x* = 0.10 obtained at temperature 700 K.

**Figure 7 materials-15-06308-f007:**
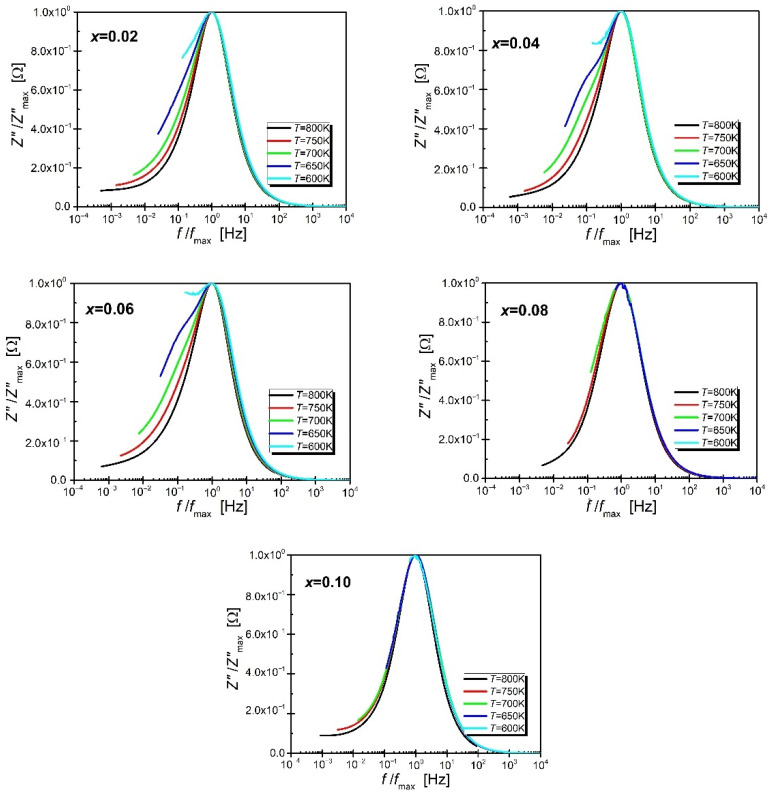
The normalized imaginary part of the impedance Z″/Z″_max_ of the BBN ceramic presented as a function of the normalized frequency f/f_max_.

**Figure 8 materials-15-06308-f008:**
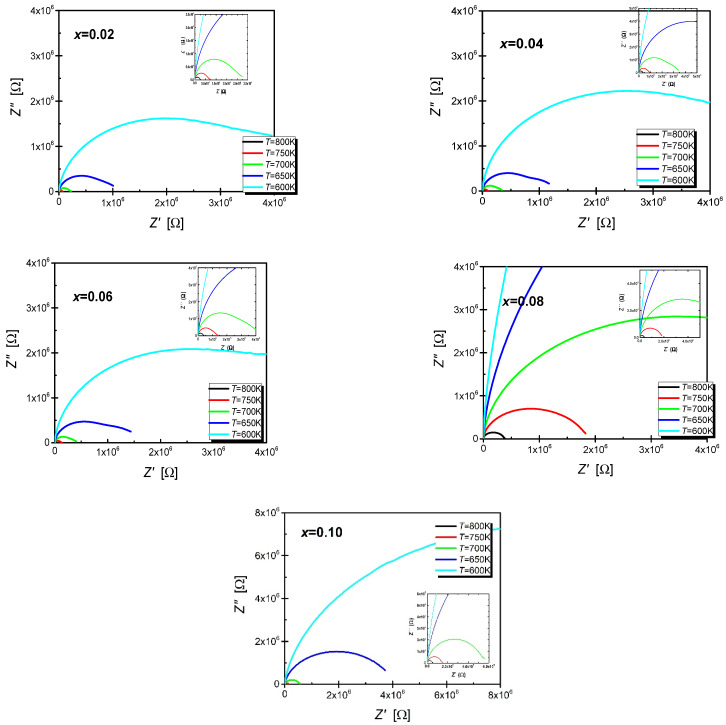
The graph of the dependence of the imaginary part of the impedance (Z″) on its real part (Z′) for the BBN ceramics doped with praseodymium, made at several selected temperatures in the range T = (500–823) K.

**Figure 9 materials-15-06308-f009:**
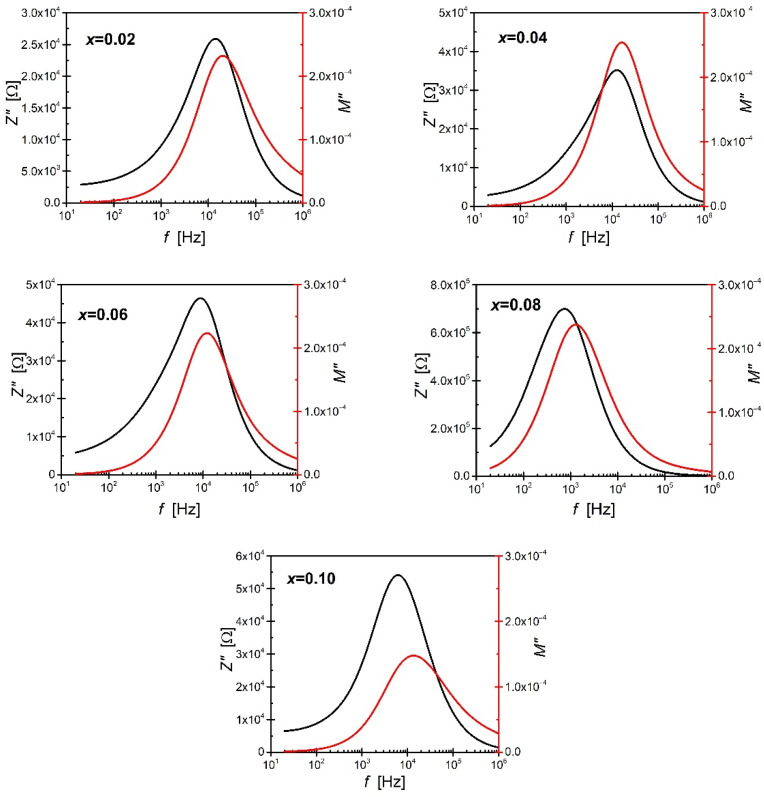
The frequency characteristics of the imaginary part of the impedance and the electric modulus correspond to the temperature for different concentrations of the Pr^3+^ admixture.

**Figure 10 materials-15-06308-f010:**
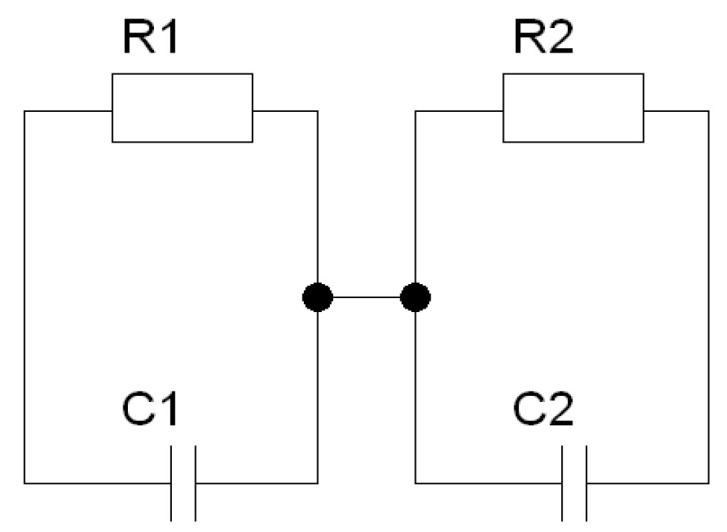
A circuit consists of two classic Voigt elements connected in series.

**Figure 11 materials-15-06308-f011:**
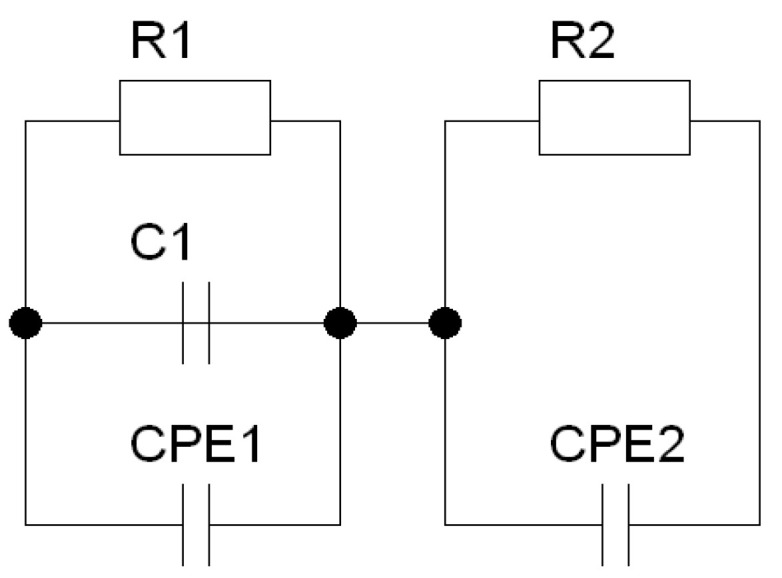
A circuit consists of two modified Voigt elements connected in series.

**Figure 12 materials-15-06308-f012:**
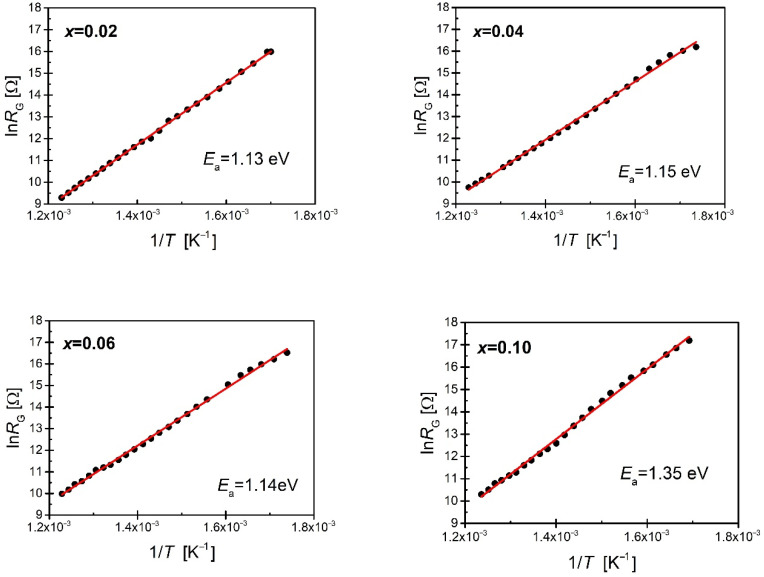
Dependence of the lnRG obtained from the analysis of impedance spectra as a function of the reciprocal of temperature.

**Figure 13 materials-15-06308-f013:**
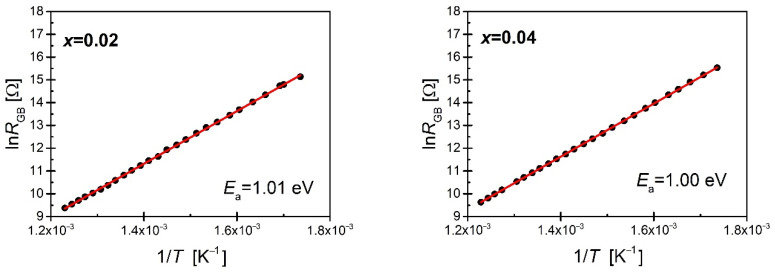
Dependence of the lnRGB obtained from the analysis of impedance spectra as a function of reciprocal of temperature.

**Figure 14 materials-15-06308-f014:**
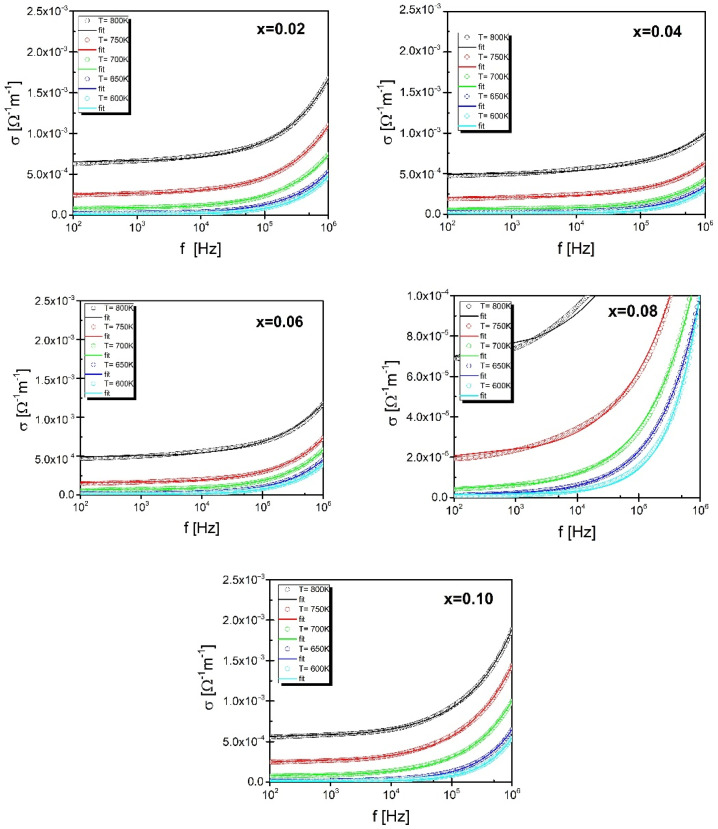
Frequency dependence of the conductivity of BBN ceramics doped with praseodymium.

**Figure 15 materials-15-06308-f015:**
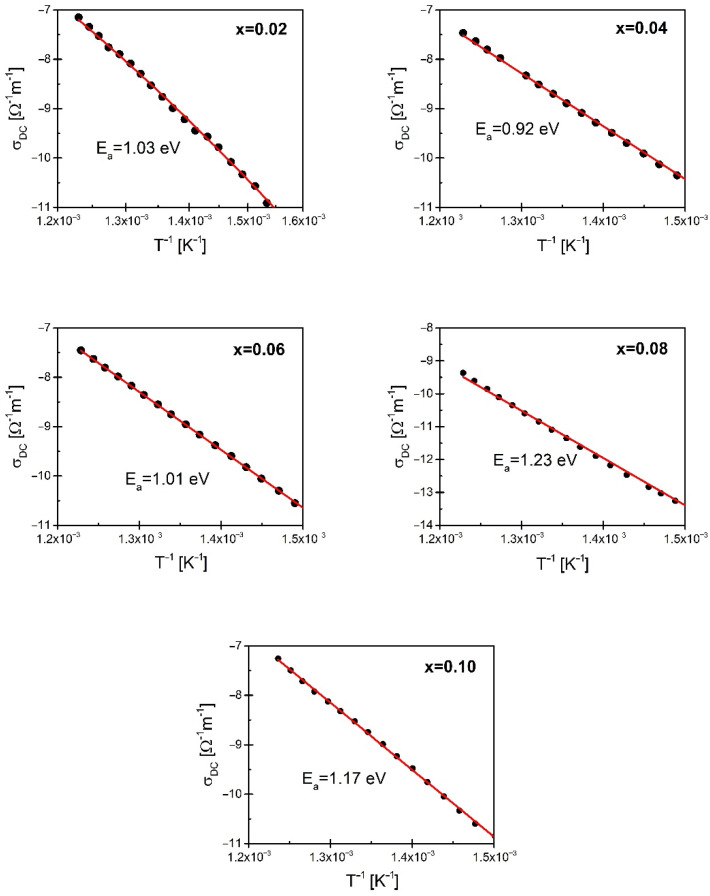
The dependence of DC conductivity versus the inverse of temperature plotted for pure and praseodymium modified and BBN ceramics.

**Figure 16 materials-15-06308-f016:**
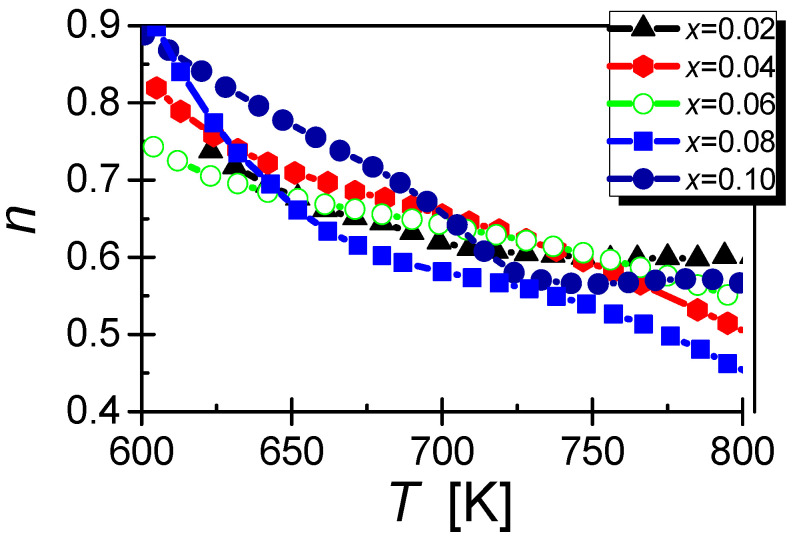
Temperature dependence of the exponent n (Joncher’s law) for BBN ceramics with modified praseodymium ions.

**Table 1 materials-15-06308-t001:** Parameters of the components of the electric circuit used for the impedance response of the BBN ceramics doped with Pr^3+^ ions in the A subnetwork at temperatures T = 600 K and T = 800 K.

Element	Parameter	*T* = 800 K				
		*x* = 0.02	*x* = 0.04	*x* = 0.06	*x* = 0.08	*x* = 0.10
**R1**	Value [Ω]	13,613	20,446	26,273	314,920	36,673
	Relative error [Ω]	58.67	178	491	1693	419
	Absolute error [%]	0.43	0.87	1.22	0.54	1.14
**CPE-T**	Value [F]	5.99 × 10^−7^	3.64 × 10^−7^	1.26 × 10^−7^	2.38 × 10^−8^	1.43 × 10^−8^
	Relative error [F]	2.31 × 10^−8^	1.02 × 10^−8^	7.81 × 10^−13^	9.11 × 10^−10^	5.36 × 10^−10^
	Absolute error [%]	3.87	2.80	0.41	3.83	3.73
**CPE-P**	Value [a.u.]	0.38	0.39	0.38	0.38	0.5451
	Relative error [a.u.]	0.0032	0.0029	0.008	0.0031	0.0022
	Absolute error [%]	0.84	0.74	6.2	0.83	0.41
**C**	Value [F]	2.88 × 10^−10^	3.27 × 10^−10^	1.25 × 10^−10^	9.06 × 10^−11^	9.921 × 10^−11^
	Relative error [F]	61.55	5.85 × 10^−12^	5.16 × 10^−13^	3.97 × 10^−13^	3.27 × 10^−13^
	Absolute error [%]	0.70	1.79	0.41	0.44	0.33
**R2**	Value [Ω]	13,989	18,166	23,729	79,871	22,069
	Relative error [Ω]	61.88	169	416	1850	452
	Absolute error [%]	0.44	0.93	1.75	2.32	2.04
**CPE-T**	Value [F]	4.196 × 10^−10^	2.48 × 10^−10^	4.74 × 10^−11^	1.14 × 10^−9^	3.47 × 10^−10^
	Relative error [F]	4.75 × 10^−12^	2.08 × 10^−12^	1.44 × 10^−11^	2.51 × 10^−11^	6.37 × 10^−12^
	Absolute error [%]	1.13	0.84	30	2.20	1.84
**CPE-P**	Value [a.u.]	0.96	0.99	0.97	0.98	0.97
	Relative error [a.u.]	0.0007	0.0006	0.04	0.002	0.004
	Absolute error [%]	0.08	0.06	4.12	0.18	0.41
	χ^2^	5.5 × 10^−6^	1.71 × 10^−5^	9.7 × 10^−5^	1.6 × 10^−5^	2.45 × 10^−5^
**Element**	**Parameter**	***T* = 600 K**				
		*x* = 0.02	*x* = 0.04	*x* = 0.06	*x* = 0.08	*x* = 0.10
**R1**	Value [Ω]	5.13 × 10^6^	5.33 × 10^6^	6.7743 × 10^6^	2.6679 × 10^7^	2.0817 × 10^7^
	Relative error [Ω]	494,494	95,650	91,936	1.24 × 10^5^	1.26 × 10^5^
	Absolute error [%]	0.97	1.79	1.36	0.45	0.61
**CPE-T**	Value [F]	2.86 × 10^−8^	3.18 × 10^−8^	1.47 × 10^−8^	6.9 × 10^−10^	1.42 × 10^−9^
	Relative error [F]	6.44 × 10^−10^	9.21 × 10^−10^	4.24 × 10^−10^	1.62 × 10^−11^	3.16 × 10^−11^
	Absolute error [%]	2.25	2.90	2.87	2.34	2.22
**CPE-P**	Value [a.u.]	0.41	0.46	0.52441	0.54	0.5874
	Relative error [a.u.]	0.003	0.0044	0.0041	0.004	0.0028
	Absolute error [%]	0.73	0.97	0.79	0.66	0.47
**C**	Value [F]	3.48 × 10^−10^	4.80 × 10^−10^	3.89 × 10^−10^	1.05 × 10^−10^	1.372 × 10^−10^
	Relative error [F]	1.17	2.63 × 10^−12^	2.44 × 10^−12^	4.68 × 10^−13^	8.59 × 10^−13^
	Absolute error [%]	0.34	0.55	0.63	0.44	0.62
**R2**	Value [Ω]	1.6833 × 10^6^	2.1603 × 10^6^	2.4574 × 10^6^	1.2728 × 10^6^	1.5381 × 10^6^
	Relative error [Ω]	7437	6414	9826	1.09 × 10^4^	1.45 × 10^4^
	Absolute error [%]	0.44	0.30	0.4	0.86	0.92
**CPE-T**	Value [F]	5.60 × 10^−10^	3.14 × 10^−10^	3.74 × 10^−10^	2.84 × 10^−8^	5.636 × 10^−8^
	Relative error [F]	3.73 × 10^−12^	1.48 × 10^−12^	2.15 × 10^−12^	1.10 × 10^−10^	2.68 × 10^−10^
	Absolute error [%]	0.67	0.47	0.58	0.38	0.48
**CPE-P**	Value [a.u.]	0.96	0.99	0.98015	0.89929	0.81243
	Relative error [a.u.]	0.0003	0.0003	0.0005	0.031	0.023
	Absolute error [%]	0.036	0.031	0.049	3.41	2.77
	χ^2^	1.35 × 10^−5^	4.55 × 10^−5^	6.67 × 10^−5^	1.9 × 10^−5^	2.41 × 10^−5^

**Table 2 materials-15-06308-t002:** Values of the activation energy of the conductivity process in grains and grain boundaries.

*x* (Pr)	*E_G_* [eV]	*E_GB_* [eV]
0.02	1.13	1.00
0.04	1.15	1.00
0.06	1.14	0.98
0.08	1.44	0.95
0.10	1.35	0.85

## Data Availability

Not applicable.

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
