# Peer review of "Impedance Spectroscopy of Pr-Doped BaBi2Nb2O9 Aurivillius Ceramics"

_materials, 2022, doi:10.3390/ma15186308_

Round 1
Reviewer 1 Report
The suggested comments to the authors
The authors have addressed the Impedance Spectroscopy of Pr-doped BaBi2Nb2O9 Aurivillius ceramics in this work. After carefully reviewing the manuscript, I noticed that the authors must go through the following suggestions carefully before taking any decision about the manuscript.
1. Better define the novelty and importance of the work. Be clear in stating what aspects of this work are new, what results were interesting or surprising and how this work will impact future research and development in this field.
2. The article lacks the citation of the new published research on this subject where there are many articles were reported in this field. The authors are requested to take the cite from new published research on the subject of the related research to account for the article to be valuable.
3. I suggested to re-write the abstract as “This work reports the polycrystalline of the Aurivilius type structure, mainly the Pr3+-doped Ba2+- site of BaBi2Nb2O9 (BBN) ceramics. The ceramics under investigation were fabricated via the conventional solid-state reaction method. Scanning electron microscopy (SEM) and energy dispersion spectroscopy (EDS) techniques were used to analyzed their morphological structure. The impedance spectroscopy technique was used to investigate the role of the applied modifications in the conductivity processes in the studied material.”
4. In Keywords: please start with capital letter such as Ceramics; Aurivillius structure; BaBi2Nb2O9; Praseodymium Pr3+, Impedance spectroscopy
5. I suggested to re-write the first paragraph of introduction as the following with cited the related literature “A group of perovskite-type materials can be described by the general formula ABX3 [1]. Two pyramids form the theoretical, undisturbed crystal structure of this kind, with a common base occupying the cube's centre [Ref]. This cell is far from ideal because there are deformations in it as a result of stresses in the structure [Ref]. They result from the difference in the ionic radii of the elements that make up a given compound. It is these deformations that determine the physical properties of the material. The commonly known representatives of perovskite electroceramics are PbZrO3-PbTiO3 [2,3] or PbFe1/2Nb1/2O3 [4,5]. The structure of the perovskite is the starting form for the construction of compounds with the structure of layered perovskites; such structures include the Aurivillius compounds, which are characterized by the general formula [6]:”
6. Barium bismuth (III) niobate, BaBi2Nb2O9 (BBN), is a Bi-layered pseudo-perovskite Aurivillius crystal having microscopic orthorhombic distortions in a macro tetragonal symmetry domain leading to a relaxor ferroelectric nature with exhibiting high spontaneous polarization [11,12].
7. The introduction should be improved with comparing this study with literature reported in bismuth layered structured? I suggested to authors to take help of the following litratures
1) Ceramics International, 40(10), 16365-16369.
2) Ferroelectrics Letters Section 43, no. 4-6 (2015): 82-89.
3) Processing and Application of Ceramics 12.1 (2018): 72-77.
8. The Authors doped Pr3+ in Ba2+ why they select this site instead of Bi3+ sit, however It is expected that by substituting Bi3+ ion with the large difference in the eightfold coordination ionic radii of Pr3+ ion in the crystal lattice of BBN relaxor ferroelectrics could enhance its physical properties.
9. How the authors decide that the Pr going to Ba site without doing XRD study?
10. The authors in P.3 L.79 mentioned that “The BBN and of BPBN ceramics were subjected to microstructural tests using a scanning electron microscope” But there are no images of pure BBN the authors asked to include it for better compression.
11. The authors should write “3. Results and discussion” not only Results
12. The authors mentioned that “The images of the microstructure of BBN ceramics doped with Pr3+ ions indicate that 86 an increase in the dopant content causes a slight reduction in the grain size”. The authors asked to calculate the grain size to support this statement? Also, it is appearing that the with x=0.10 having large grain size.
13. Please revise the figure numbers in the figures captions as well as in text in the entire of manuscript except Figure 1; all are wrong
14. Figure 2. The frequency characteristics of the imaginary part of the impedance and the electric module corresponding to the temperature for different concentrations of the Pr3+ admixture please revise these errors
15. Figure 3. The linear nature of the relationship indicates the activating form of conductivity processes. Based on the Arrhe- 206 nius relationship (7), the activation energy of the conduction process of both EG grains and EGB grain boundaries was 207 determined (Table 3).
16. Fig. 1 Frequency dependence of DC conductivity on T-1 temperature of BaBi2Nb2O9 ceramics doped with praseodymium.
17. Figure 6. Temperature dependence of the exponent n (Joncher's law) for BaBi2Nb2O9 ceramics with modified praseodymium ions.
18. Please revise the Table numbers in the captions and in the text.
Reviewer 2 Report
In the review of the manuscript titled: Impedance Spectroscopy of Pr-doped BaBi2Nb2O9 Aurivillius ceramics. The authors have provided a poor description and possess no clear information about this material. In my opinion, I would reject this article as it requires so much modification. My few suggestions for the authors to improve the manuscript are as follow,;
1. i- Abstract of the work is very weak as compared to results. Readers know that material belongs to a Aurivillius BLSF’s material, it’s so basic for abstract, kindly remove it. Moreover, English must be modified throughout the manuscript i.e.,
ii- Page 1, line 11, “The test of material was produced….” Is it a test or fabrication technique, I don’t think so this basic information is necessary in abstract or conclusion?
iii- Spelling correction, page 1 line 10 “Aurivillius”
iv- Authors are advised to highlight their achieved results in the abstract rather than the fabrication or characterization information.
2. In the section of Introduction Page 1, line 17, there is no logic of such sentence in the first paragraph “There is a group of perovskite-type materials that can be described by the general formula ABX3”.
3. Introduction paragraph 1 is so much basic and no clearance of concepts by authors.
4. In the paragraph 2 authors state “The interest in BBN ceramics appeared at the beginning of the 21st century. However, to this day, there is an insufficient amount of information in the world literature on how to modify it with dopant ions, particularly with ions of rare earth elements.” I do not agree with the authors, as there is much research going on these materials, more than doping and additions various authors are working on the new concepts about these materials. Please correct your information e.g.,
J Asian Ceram Soc. 2021;9(1):312–322
J Am Ceram Soc. 2017;100(8):3522–3529.
J Alloys Compd. 2018;740:1–6.
Materials Technology, DOI: 10.1080/10667857.2022.2059046
5. Introduction portion is weak as only basic information is described; there must be detailed literature review of the material.
6. Problem statement is missing, why are the authors going to perform this research work, what was their motivation?
7. Page 2, Line 45, “The aim of the work was the technology of BBN ceramics modified….”, Which technology? There is no sense of this sentence, please make it clear.
8. In my opinion the microstructure of BLSF’s must be plate like from top side and cross-section wise needle like. Herein, there is a grain like or particle like structure. I think dopant has influenced the morphology. I would like to see the pure BBN micrograph in Figure 1, it will provide the clear comparison. In short I am not satisfied with the result.
9. XRD analysis for the material fabrication is always necessary tool, we can never say that on the basis of SEM or EDS we have successfully fabricated our material. Hence for the structural confirmation and phase structure of effect of dopant on the lattice sites, XRD is important. Please add the result.
10. In the EDS diagram, atomic weight ratio of elements is missing; kindly add it for stoichiometric ratio od added elements.
11. LCR meter dependent results are not as aspiring, which are average for this material. It will be better to add some electrical, ferroelectric, piezoelectric or magnetic properties of the material to make the material attractive for the applications.
Reviewer 3 Report
The manuscript entitled “Impedance Spectroscopy of Pr-doped BaBi2Nb2O9 Aurivillius ceramics” by Rerak et al describes the synthesis of different Pr doped 2-layered ceramics of Aurvillius family. They synthesized ceramic samples of BBN by the solid-state synthesis. Detailed relaxation studies by Z and M analysis is presented. Jonscher relations for different doped ceramics have been presented. Authors have nicely presented their fitting results of impedance analysis. The work looks interesting for the readers of Materials journal.
However, authors haven’t pointed out important research findings in a presentable manner.
In the abstract they should mention structural aspects of the BBN samples when doped with Pr. SEM-EDS results are not enough to confirm the sample.
The manuscript should be presented in more ordered form. For example:
(1) The introduction should present why authors have chosen this material?what is the main aim of the investigation?what do they expect from the study.
(2) An improved introduction should be mentioned presenting the importance of BBN, like the material is of relaxor ferroelectric nature, low-loss dielectric etc.
(3) If the main aim of the investigation is only to study impedance spectroscopy, they should mention if any such literature is available either on BBN or analogous 2-layered samples?
(4) They synthesized the ceramics by a solid-state procedure. The same should be mentioned in the abstract but not solid-phase synthesis or technology or free air sintering etc. The technical terminology should be followed according to the literature or article published in Materials journal.
(5) Why the authors did not mention the x-ray diffraction of the samples? To confirm the expected phase XRD images should be presented with miller indices matched to the reference sample.
(6) What is the density of the ceramics?
(7) From the SEM analysis what do they infer? As a function of Pr is there any change in microstructure?in terms of grain size, shape and phases.
(8) Aurivillius oxides are generally characterized from their morphology through SEM which normally display needle or platelets. Do they have any such analysis?
(9) Lattice parameters are important parameters to understand whether Pr is replaced in some Bi atoms. Such analysis should be presented.
(10) Since the microstructure plays an important factor in the conductivity of BBN after Pr doping, the effect of Pr on grain growth and impedance analysis should be mentioned/discussed with some correlations with literature of either 2-layer or other Aurivillius oxides.
(11)As mentioned in the earlier points grain and grain boundary resistance and oxygen vacancies are Pr doping dependent, they should be discussed.
(12)BBN is a ferroelectric material; do they have any phase transition results from the dielectric analysis?
(13)The terminology of the manuscript should be formatted as per the research articles but not like the text book.
Based on the above missing details and formatting errors, the manuscript can not be accepted in the present version. Hence I suggest a major revision. This also gives the authors another chance to revise the manuscript by considering above issues.
Round 2
Reviewer 1 Report
Indeed, the authors considered the suggested comments, which improved the manuscript's scientific value.
Author Response
Dear Reviewer,The authors would like to thank the Reviewer once again for valuable comments and tips.
The native speaker corrected the article. Linguistic and stylistic errors have been eliminated.
Yours faithfully,
Authors
Reviewer 2 Report
In the second review report of the article titled: Impedance Spectroscopy of Pr-doped BaBi2Nb2O9 Aurivillius ceramics, authors have described they have revised the article under the suggestions of the reviewers. Authors have described the manuscript better but still much improvement is required, especially the English should be revised again. I would like to recommend the article publish but requires minor modification.
1. Abstract: page 1, line 9, “This work reports the polycrystalline of the Aurivillius type structure, mainly the Pr3+- doped Ba2+- site of BaBi2Nb2O9 (BBN) ceramics”, sentence makes no sense at all. Please correct it. Herein this study, the polycrystalline nature of the Aurivillius type structure is studied, primarily the main objective is to observe the influence of dopant Pr3+ at the Ba2+-site of BaBi2Nb2O9 (BBN) ceramics.
2. Abstract: page 1, line 12, “were used to analyzed”, please correct the grammar.
3. Introduction: Page 1, line 25, I mentioned in report 1, here its same again, “A group of perovskite-type materials can be described by the general formula ABX3 [1]”, do this sentence has any reason? Any concept?, please combine first two sentences togather to make the clear presentation.
4. Please mention the peak index using the reported PDF card number to confirm the fabrication of single phase BaBi2Nb2O9 Aurivillius ceramics.
5. Page 4, line 124, Please check the sentence, “were compared with the one from the international database” Which international debate?
6. How authors have calculated these values of unit volume? “The unit cell volume equals 398.1 and 395.8 for pure BBN ceramic”.
